# Tightest Admissible Shortest Path

## Abstract

The shortest path problem in graphs is fundamental to AI. Nearly all variants of the problem and relevant algorithms that solve them ignore edge-weight computation time and its common relation to weight uncertainty. This implies that taking these factors into consideration can potentially lead to a performance boost in relevant applications. Recently, a generalized framework for weighted directed graphs was suggested, where edge-weight can be computed (estimated) multiple times, at increasing accuracy and run-time expense. We build on this framework to introduce the problem of finding the tightest admissible shortest path (TASP); a path with the tightest suboptimality bound on the optimal cost. This is a generalization of the shortest path problem to bounded uncertainty, where edge-weight uncertainty can be traded for computational cost. We present a complete algorithm for solving TASP, with guarantees on solution quality. Empirical evaluation supports the effectiveness of this approach.

## 1 Introduction

Finding the shortest path in a directed, weighted graph is fundamental to artificial intelligence and its applications. The *cost* of a path is the sum of the weights of its edges. Informed and uninformed search algorithms for finding *shortest* (minimal-cost) paths are heavily used in planning, scheduling, machine learning, constraint optimization, etc.

Graph edge-weights are commonly assumed to be available in negligible time. However, this does not hold in many applications. When weights are determined by queries to remote sources, or when a massive graph is stored in external memory (e.g., disk) then the order in which edges are visited—accessing external memory—needs to be optimized (Vitter 2001; Hutchinson, Maheshwari, and Zeh 2003; Jabbar 2008; Korf 2008b,a, 2016; Sturtevant and Chen 2016). Similarly, when edge-weights are computed dynamically using learned models, or external procedures, it is beneficial to delay weight evaluation until necessary (Dellin and Srinivasa 2016; Narayanan and Likhachev 2017; Mandalika, Salzman, and Srinivasa 2018; Mandalika et al. 2019).

Instead of delaying and re-ordering expensive edge-weight evaluations, a recent formalization focuses instead on using multiple *weight estimators* (Weiss and Kaminka 2023b; Weiss, Felner, and Kaminka 2023). Edge-weights are replaced with an ordered set of estimators, each providing lower and upper bounds on the true weight. Incrementally, subsequent estimators can tighten the bounds, but at increasing computation time. A search algorithm may quickly compute loose bounds on the edge-weight, and invest more computation on a tighter estimator later in the process.

For example, consider finding the fastest route between two cities, where edges and their weights represent roads and travel times, resp. First rough bounds on travel time can be estimated from fixed distances and speed limits (say, from a local database). For more accuracy, Google Maps, which considers additional road factors and traffic data, can be queried online. Even more accuracy can be achieved—at increased run-time—by further considering *vehicle attributes* together with road characteristics, which can be highly relevant for large and heavy vehicles such as trucks and buses (PTV-Group 2023).

Having multiple weight estimators for edges is a proper generalization of standard edge-weights, and raises several shortest path problem variants. The classic singular edge-weight is a special case, of an estimator whose lower- and upper- bounds are identical. However, since the true weight may not be known (even applying the most expensive estimator), other variants of the shortest path problems involve finding paths that have the best bounds on the optimal cost.

In this paper, we introduce the *tightest admissible shortest path* (TASP) problem, which is an important shortest-path problem variant in graphs with multiple-estimated edge-weights. Its solution provides an answer to the question: what is the tightest suboptimality factor w.r.t. the optimal cost that can be achieved given that edge-weights have uncertainty bounds? We show that solving TASP can be reduced to solving two simpler problems: The *shortest path tightest lower bound* (SLB), that was introduced and solved optimally by the BEAUTY algorithm in (Weiss, Felner, and Kaminka 2023); and the *shortest path tightest upper bound* (SUB) problem, which we define and solve in this paper.

To solve SUB, we present BEAST, an uninformed search algorithm based on *uniform-cost search* (UCS, a variant of *Dijksra's* algorithm) (Dijkstra 1959; Felner 2011), that takes advantage of prior information about the solution of SUB. We then use it to construct BEAUTY&BEAST, an algorithm that solves TASP problems by exploiting coupling between the problems of SLB and SUB. Experiments demonstrate the effectiveness of BEAUTY&BEAST compared to the base-

line that solves SLB and SUB separately.

Our main contributions in this paper are threefold: **(1)** The introduction of TASP as a natural extension of the familiar shortest path problem to settings with bounded edge-weight uncertainty, and its solution by solving the related problems SLB&SUB. **(2)** The BEAST algorithm that finds optimal solutions for SUB. **(3)** The idea that information obtained from solving either SLB or SUB can be used to enhance the solution of the other (which is demonstrated by BEAUTY&BEAST).

## 2 Background and Related Work

This paper belongs to a line of works that consider the run-time of edge-evaluation to be non-negligible, within the context of graph search. It is thus useful to consider the overall edge-evaluation time $T_e$ and its decomposition as $T_e = \tau_e \times n_e$, where $\tau_e$ is the average edge-evaluation time and $n_e$ is the total number of edge-weight computations conducted over all edges throughout the search. In contrast to standard search algorithms that assume that $\tau_e$ is negligible and thus focus on pure search time, algorithms in the setting discussed in this paper utilize various techniques in order to reduce $T_e$, sometimes at the expanse of increased search effort (e.g., more node expansions), with the aim to minimize overall run-time.

In the domain of robotics it is common to have shortest path problems in robot configuration spaces, where $\tau_e$ is typically *high*, as in these applications edge existence and cost are determined by expensive computations for validating geometric and kinematic constraints. A well known technique in these cases is to reduce $n_e$ by explicitly delaying weight computations (Dellin and Srinivasa 2016; Narayanan and Likhachev 2017; Mandalika, Salzman, and Srinivasa 2018; Mandalika et al. 2019), even at the cost of increasing search effort. Related challenges arise in planning, where action costs can be computed by external (potentially expensive-to-compute) procedures (Dornhege et al. 2012; Gregory et al. 2012; Frances et al. 2017), or when multiple heuristics have different run-times (Karpas et al. 2018).

There are also approaches that aim to reduce $\tau_e$, instead of $n_e$. When the graph is too large to fit in random-access memory, it is stored externally (i.e., disk). External-memory graph search algorithms optimize the memory access patterns for edges and nodes, to make better use of faster memory (caching) (Vitter 2001; Hutchinson, Maheshwari, and Zeh 2003; Jabbar 2008; Korf 2008b,a, 2016; Sturtevant and Chen 2016). This reduces $\tau_e$ by amortizing the computation costs, but still assumes a single computation per edge.

The approach we take in this paper follows a recent line of work—that is complementary to those described above—which focuses on using multiple *edge estimators* (Weiss and Kaminka 2023b; Weiss, Felner, and Kaminka 2023), specifically for estimating edge-weights. In this framework the weight of each edge can be estimated multiple times, successively, at increasing expense for greater accuracy. The idea is that in some cases cheaper weight estimates can be used instead of the best and most expensive estimates, thus decreasing $\tau_e$, and although $n_e$ might increase, the overall edge-evaluation time $T_e$ might still decrease.

Lastly, there are also works that consider weight uncertainty in graphs, regardless of edge-computation time. These include, e.g., the case where weights are assumed to be drawn from probability distributions (Frank 1969), and the usage of fuzzy weights (Okada and Gen 1994) that allow quantification of uncertainty by grouping approximate weight ranges to several representative sets. All these lines of work ignore the weight *computation time*, in contrast to the work reported here.

## 3 Shortest Path with Estimated Weights

**Graph Definitions.** A *weighted digraph* is a tuple $(V, E, c)$, where $V$ is a set of nodes, $E$ is a set of edges, s.t. $e = (v_i, v_j) \in E$ iff there exists an edge from $v_i$ to $v_j$, and $c : E \to \mathbb{R}^+$ is a cost (weight) function mapping each edge to a non-negative number. Let $v_i$ and $v_j$ be two nodes in $V$. A *path* $\pi = \langle e_1, \ldots, e_n \rangle$ from $v_i$ to $v_j$ is a sequence of edges $e_k = (v_{q_k}, v_{q_{k+1}})$ s.t. $k \in [1, n]$, $v_i = v_{q_1}$, and $v_j = v_{q_{n+1}}$. The cost of a path $\pi$ is then defined to be $c(\pi) := \sum_{k=1}^{n} c(e_k)$. The *Goal-Directed Single-Source Shortest Path* ($GDS^3P$) problem involves finding a *solution*, which is a path $\pi$ from the source node to a goal node, with minimal $c(\pi)$, denoted as $C^*$. A solution $\pi$ for a $GDS^3P$ problem is said to be a $\mathcal{B}$-**admissible shortest path** if $c(\pi)$ is bounded by a suboptimality factor $\mathcal{B}$, i.e.,

$$c(\pi) \leq C^* \times \mathcal{B}. \tag{1}$$

If $\mathcal{B} = 1$, then $\pi$ is a shortest path.

We recall several definitions that were introduced in (Weiss, Felner, and Kaminka 2023).

**Definition 1.** A **cost estimators function** $\Theta$, *for a set of edges $E$ and with a cost function $c : E \to \mathbb{R}^+$, maps every edge $e \in E$ to a finite and non-empty sequence of* weight estimation procedures,

$$\Theta(e) := (\theta_e^1, \ldots, \theta_e^{k(e)}), k(e) \in \mathbb{N}, \tag{2}$$

*where* **estimator** $\theta_e^i$, *if applied, returns lower- and upper-bounds $(l_e^i, u_e^i)$ on $c(e)$, such that $0 \leq l_e^i \leq c(e) \leq u_e^i < \infty$). $\Theta(e)$ is ordered by the increasing running time of $\theta_e^i$, and the bounds monotonically tighten, i.e., $[l_e^j, u_e^j] \subseteq [l_e^i, u_e^i]$ for all $i < j$.*

**Definition 2.** *An* **estimated weighted digraph (EWDG)** *is a tuple $G = (V, E, c, \Theta)$, where $V, E$ are sets of nodes and edges, resp., $c$ is an* un-observable *cost function for the edges in $E$, and $\Theta$ is a* **cost estimators function** *for $E$.*

**Definition 3.** *For an edge $e$, the* **tightest edge lower bound** *and* **tightest edge upper bound** *w.r.t. $\Theta$ are $l_{\Theta(e)} := l_e^{k(e)}, u_{\Theta(e)} := u_e^{k(e)}$. For a path $\pi$, the* **tightest path lower bound** *and* **tightest path upper bound** *w.r.t. $\Theta$ follow, respectively, from the tightest edge bounds defined above.*

$$l_{\Theta(\pi)} := \sum_{i=1}^{n} l_{\Theta(e_i)}, \quad u_{\Theta(\pi)} := \sum_{i=1}^{n} u_{\Theta(e_i)} \tag{3}$$

Intuitively, $l_{\Theta(\pi)}$ and $u_{\Theta(\pi)}$ are the best estimations that are provided by $\Theta$ for the true cost of the path $\pi$, and they satisfy $l_{\Theta(\pi)} \leq c(\pi) \leq u_{\Theta(\pi)}$.

**Shortest Path Problems.** Regular weighted digraphs are a special case of EWDGs where for every edge $e$, there is a single estimation procedure $\theta_e^1 = (c(e), c(e))$ with lower and upper bounds that are identical to the weight $c(e)$. In this special case, a shortest tightly-bounded path $\pi$ in the graph is an optimal solution for a $GDS^3P$ problem. However, in the general case of EWDGs, multiple estimators exist per edge, and we are not guaranteed that every weight can be estimated precisely, even if all estimators for it are used, as the best estimator available for an edge can still deviate from the exact true cost provided by $c$. Thus, several variants of the shortest path problem exist, which correspond to different tightest bounds for the shortest path.

In this paper we focus on the problem of finding the smallest possible suboptimality factor on the optimal cost $C^*$ as well as finding a path $\pi$ that achieves it, in a given EWDG. In order to properly define this problem, we first have to extend the notion of $\mathcal{B}$-admissibility to EWDGs. Indeed, Inequality (1), which characterizes the suboptimality factor, requires knowing exact edge costs to obtain $c(\pi)$ and $C^*$, but the cost function $c$ is un-observable in EWDGs. Instead, we introduce an extension (Def. 4) that uses edge cost estimates that are provided by $\Theta$. The extension relies on $L^*$, which is the best lower bound that can be derived for $C^*$, that was introduced in (Weiss, Felner, and Kaminka 2023):

$$L^* := \min_{\pi'}\{l_{\Theta(\pi')} \mid \pi' \text{ is a path from } v_s \text{ to } v \in V_g\}. \quad (4)$$

**Definition 4.** *Let $P = (G, v_s, V_g)$, where $G$ is an EWDG with cost estimators function $\Theta$, $v_s \in V$ is the source node and $V_g \subset V$ is a set of goal nodes. A solution $\pi$ is said to be a $\mathcal{B}$-admissible shortest path w.r.t. $\Theta$ if the following holds*

$$u_{\Theta(\pi)} \leq L^* \times \mathcal{B}. \quad (5)$$

Note that $u_{\Theta(\pi)}$ is the tightest upper bound for $c(\pi)$ w.r.t. $\Theta$ (Eq. (3)), so $c(\pi) \leq u_{\Theta(\pi)}$ holds. Similarly, $L^*$ is the tightest lower bound for $C^*$ w.r.t. $\Theta$ (Eq. (4)), so $L^* \leq C^*$ holds. Thus, if Inequality (5) holds, then

$$c(\pi) \leq u_{\Theta(\pi)} \leq L^* \times \mathcal{B} \leq C^* \times \mathcal{B} \quad (6)$$

is necessarily satisfied. Namely, standard $\mathcal{B}$-admissibility is assured by $\mathcal{B}$-admissibility w.r.t. $\Theta$.

Next, we define the main problem addressed in this paper.

**Problem 1** (TASP, finding $\mathcal{B}^*$). *Let $P = (G, v_s, V_g)$, where $G$ is an EWDG with cost estimators function $\Theta$, $v_s \in V$ is the source node and $V_g \subset V$ is a set of goal nodes. Let $\mathcal{B}^*$ be the tightest $\mathcal{B}$-admissibility factor w.r.t. $\Theta$, i.e.,*

$$\mathcal{B}^* := \min_{\pi'}\{\mathcal{B} \mid u_{\Theta(\pi')} \leq L^* \times \mathcal{B}, \pi' \text{ is a solution}\}. \quad (7)$$

*The **Tightest Admissible Shortest Path** (TASP) problem is to find $\mathcal{B}^*$ as well as a solution $\pi$ that its $\mathcal{B}$-admissibility factor is $\mathcal{B}^*$. If $\mathcal{B}$-admissibility cannot be obtained for any finite value of $\mathcal{B}$, or no solution exists, then $\mathcal{B}^* = \infty$ should be returned.*

Next, we describe two problems that are related to Prob. 1. The first problem among the two deals with finding a shortest path w.r.t. lower bounds. It was introduced in (Weiss, Felner, and Kaminka 2023), and we review its definition here.

**Problem 2** (SLB, finding $L^*$). *Let $P = (G, v_s, V_g)$, where $G$ is an EWDG with cost estimators function $\Theta$, $v_s \in V$ is the source node and $V_g \subset V$ is a set of goal nodes. The **Shortest path tightest Lower Bound** (SLB) problem is to find a solution $\pi$, such that $\pi$ has the lowest tightest path lower bound of any path from $v_s$ to $v \in V_g$, w.r.t. $\Theta$, i.e., $l(\pi) = L^*$.*

The second problem, which we define here for the first time, is complementary to Prob. 2 and deals with finding a shortest path w.r.t. upper bounds.

**Problem 3** (SUB, finding $U^*$). *Let $P = (G, v_s, V_g)$, where $G$ is an EWDG with cost estimators function $\Theta$, $v_s \in V$ is the source node and $V_g \subset V$ is a set of goal nodes. The **Shortest path tightest Upper Bound** (SUB) problem is to find a solution $\pi$, such that $\pi$ has the lowest tightest path upper bound of any path from $v_s$ to $v \in V_g$, w.r.t. $\Theta$, i.e., $u(\pi) = U^*$, with*

$$U^* := \min_{\pi'}\{u_{\Theta(\pi')} \mid \pi' \text{ is a path from } v_s \text{ to } v \in V_g\}. \quad (8)$$

We next show that Problems 1–3 are all generalizations of standard $GDS^3P$ (see Thm. 1 below). In addition, they are also related to each other, in that their solutions are linked. Indeed, Thm. 2 below shows that the best lower bound for $C^*$ (Prob. 2) and the best upper bound for $C^*$ (Prob. 3) can be used to calculate the best suboptimality factor $\mathcal{B}^*$ (Prob. 1). Thm. 2 also shows that an optimal solution path for Prob. 3 is also an optimal solution path for Prob. 1. The proof of Thm. 2 also implies that if $U^* > L^* = 0$ it is impossible to prove $\mathcal{B}$-admissibility at all.

**Theorem 1** (Generality). *Problems 1, 2 and 3 are generalizations of a $GDS^3P$ problem.*

*Proof.* Any $GDS^3P$ problem can be formulated as Problem 1, or 2, or 3, by considering the special case where each edge has one estimator (namely, $k(e) = 1$ for every $e$), that returns the exact cost (i.e., $l_e^1 = c(e) = u_e^1$). In this special case it holds that $L^* = C^* = U^*$ and $\mathcal{B}^* = 1$ (in the case of $L^* = U^*$ we set $\mathcal{B}^* = 1$ even if $C^* = 0$ as there is no uncertainty at all). Therefore, the cost of the solutions of these problems are all equal to the minimum cost $C^*$, hence are by definition shortest paths. $\square$

**Theorem 2** ($\mathcal{B}^* = U^*/L^*$). *Let $P = (G, v_s, V_g)$, where $G$ is an EWDG with cost estimators function $\Theta$, $v_s \in V$ is the source node and $V_g \subset V$ is a set of goal nodes. If $L^* > 0$ for $P$, then a solution path $\pi$ for $P$ is a tightest admissible shortest path iff it is a shortest path tightest upper bound (i.e. a solution that achieves $U^*$). Furthermore, $\mathcal{B}^* = U^*/L^*$.*

*Proof.* Assume $L^* > 0$ for $P$. By definition (of TASP and of $\mathcal{B}^*$) a solution $\pi$ is a tightest admissible shortest path iff it achieves the lowest $\mathcal{B}$-admissibility factor, i.e., iff it satisfies

$$\pi = \arg\min_{\pi'} u_{\Theta(\pi')}/L^*. \quad (9)$$

Since $L^*$ does not change with different choices of $\pi'$ in the argmin expression of Eq. (9), it follows that

$$\pi = \arg\min_{\pi'} u_{\Theta(\pi')}. \quad (10)$$

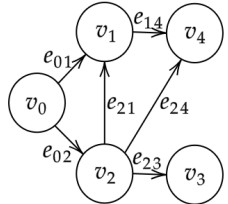

| | $e_{01}$ | $e_{02}$ | $e_{14}$ | $e_{21}$ | $e_{23}$ | $e_{24}$ |
|---|---|---|---|---|---|---|
| $c$ | 4 | 4 | 5 | 3 | 7 | 6 |
| $\theta^1$ | (4,4) | (2,6) | (1,10) | (2,3) | (7,9) | (4,6) |
| $\theta^2$ | | (3,5) | (4,6) | (3,3) | (7,8) | |

Figure 1: Left: Digraph $G$. Right: costs and estimates of $G$.

By definition of $U^*$ (Eq. (8)), a solution $\pi$ satisfying Eq. (10) achieves $u_{\Theta(\pi)} = U^*$. On the other hand, a solution satisfying Eq. (8), also achieves $\mathcal{B}^*$, as implied by Eq. (9). Thus, $\pi$ is a tightest admissible shortest path iff it is a shortest path tightest upper bound (SUB). Furthermore, it holds that $\mathcal{B}^* = U^*/L^*$. □

**Corollary 1.** *Thm. 2 shows how to obtain an optimal solution for Prob. 1 from optimal solutions for Problems 2 and 3. Thus, any algorithmic improvement to solving either Prob. 2 or Prob. 3 directly affects the efficiency of solving Prob. 1.*

Next, we use an example taken from (Weiss, Felner, and Kaminka 2023) with slight modification and supplement it to illustrate the meaning of the solutions for Problems 1–3.

**Example 1.** *Consider the estimated weighted digraph $G = (V, E, c, \Theta)$ provided in Fig. 1. Given the graph above, we may define the problem $P = (G, v_s, V_g)$ with $v_s = v_0$ and $V_g = \{v_3, v_4\}$, i.e., searching for paths from $v_0$ to either $v_3$, or $v_4$. Then, the unknown optimal cost is $C^* = c(\pi^*) = c_{01} + c_{14} = 9$ with $\pi^* = \langle e_{01}, e_{14} \rangle$; the tightest lower bound for $C^*$ is $L^* = l_{\Theta(\pi_1)} = l_{02}^2 + l_{24}^1 = 7$ with $\pi_1 = \langle e_{02}, e_{24} \rangle$ (the SLB solution); the tightest upper bound for $C^*$ is $U^* = u_{\Theta(\pi_1)} = u_{01}^1 + u_{14}^2 = 10$ with $\pi_2 = \pi^*$ (the SUB solution); and tightest admissibility factor is $\mathcal{B}^* = U^*/L^* = 10/7$ with $\pi_3 = \pi_2$ (the TASP solution).*

# 4 Algorithms

As indicated in Corollary 1, we can obtain optimal solutions for TASP problems by optimally solving the corresponding SLB and SUB problems. SLB was studied in (Weiss, Felner, and Kaminka 2023), which introduced BEAUTY, a complete algorithm that finds optimal solutions for it. Hence, this section focuses on solving SUB, particularly towards solving TASP in a manner that takes advantage of information already obtained during the solution of SLB.

In Subsection 4.1 we present BEAST (Branch&bound Estimation Applied to Searching for Top, Alg. 1), a complete algorithm that finds optimal solutions for SUB problems. BEAST extends UCS to dynamically apply cost estimators during a best-first search w.r.t. upper bounds of edge costs, and it aims to reduce the number of expensive estimators used, by utilizing both cheaper estimates and prior information on $U^*$. BEAST is thus particularly suitable to be used whenever prior information on $U^*$ is available.

We note that BEAST has several analogies to BEAUTY, but also several fundamental differences which are due to the fact that *upper bounds tighten towards lower values* whereas

*lower bounds tighten towards higher values*. Thus, cheaper (and looser) estimates cannot be used analogously to guide the search in minimization of tight upper bounds (SUB) compared to minimization of tight lower bounds (SLB). Specifically, BEAUTY first uses a looser lower bound to guarantee that a more expensive lower bound estimate is required. This can be done since if a path $\pi_1$ to a node $n$ and its tightest path lower bound $l_{\Theta(\pi_1)}$ are known, and a new path $\pi_2$ to $n$ is considered, then if a loose lower bound of $c(\pi_2)$ is already greater than $l_{\Theta(\pi_1)}$, then necessarily $l_{\Theta(\pi_2)} > l_{\Theta(\pi_1)}$ (i.e., $\pi_1$ is better). Hence, more expensive estimates for $\pi_2$ can be avoided. In contrary, this cannot be done analogously with looser upper bounds. Indeed, knowing that a loose upper bound of $c(\pi_2)$ is already greater than $u_{\Theta(\pi_1)}$ does not imply that $u_{\Theta(\pi_2)} > u_{\Theta(\pi_1)}$. Hence, more expensive estimates for $\pi_2$ cannot be avoided in the same way. For this reason, BEAST makes use of a combination of upper and lower bounds to try to avoid expensive estimates (see full description in Subsection 4.1).

In Subsection 4.2 we introduce BEAUTY&BEAST (Alg. 2), a complete algorithm that finds optimal solutions for TASP problems, that first uses BEAUTY to optimally solve SLB, and then uses BEAST, together with prior information already obtained on $U^*$, to optimally solve SUB.

## 4.1 The First Algorithm: BEAST

Algorithm 1 receives an SUB problem instance and one hyper-parameter $u_{prune}$. For simplicity we will first describe a *base case* where $u_{prune}$ is set to $\infty$, and therefore has no effect and can be ignored. The relevant instructions using $u_{prune}$ are colored in blue (Lines 12, 15) and should be ignored for now. We will come back to this parameter later.

**Base Setting.** BEAST is structurally similar to UCS. It activates a best-first search process using the standard OPEN and CLOSED lists. Nodes $n$ in OPEN are prioritized by $g_u(n)$ which is always equal to the optimal upper bound to node $n$ along the best known path (similar to using $g(n)$ for ordering nodes in UCS in regular graphs, which is done according to optimal cost). The *best* such node $n$ is chosen for expansion in Line 4, and its successors are added in the loop of Lines 8–19. When a goal node $n$ is found in Line 5, the solution path $\pi$ ending in $n$ and its tight upper bound $g_u(n) = U^*$ are returned (Thm. 3).

The main difference of BEAST over UCS is in the *duplicate detection* mechanism performed when evaluating the cost of a new edge $e$ that connects $n$ to its successor $s$. In UCS, the exact edge cost $c(e)$ is immediately obtained and used to update the path cost that ends in $s$. In BEAST, we iterate over the different estimators $\theta_e^i$ for edge $e$ (Lines 12–13). In each iteration $g_u(n) + l(e)$ serves as a lower bound for the tightest path upper bound of the path to $s$ given the current estimator (Line 13). Namely, the tight upper bound is used up to node $n$ and a lower bound is used for the edge $e$ from $n$ to $s$. Now such a path can be already pruned earlier if its current lower bound for the tight upper bound (using the current estimator) will not improve the best known path to $s$ ($g_u(s)$). In that case we will not need to further activate more expensive estimators. Thus, if $g_u(n) + l(e) \geq g_u(s)$,

**Algorithm 1: BEAST**

**Input**: Problem $P = (G, v_s, V_g)$, where $G$ is an estimated weighted digraph with cost estimators function $\Theta$
**Parameter**: Threshold $u_{prune}$
**Output**: Path $\pi$, bound $U^*$

1: $g_u(s_0) \leftarrow 0$; OPEN $\leftarrow \emptyset$; CLOSED $\leftarrow \emptyset$;
2: Insert $s_0$ into OPEN with key $g_u(s_0)$
3: **while** OPEN $\neq \emptyset$ **do**
4:     $n \leftarrow$ Pop node $n$ from OPEN with minimal $g_u(n)$
5:     **if** $Goal(n)$ **then**
6:        **return** $trace(n), g_u(n)$        // return $\pi, U^*$
7:     Insert $n$ into CLOSED
8:     **for each** successor $s$ of $n$ **do**
9:        **if** $s$ not in OPEN $\cup$ CLOSED **then**
10:           $g_u(s) \leftarrow \infty$
11:        $l(e) \leftarrow 0, u(e) \leftarrow 0$
12:        **while** $g_u(n) + l(e) < g_u(s)$ **and** $g_u(n) + l(e) \leq u_{prune}$ **and** estimators remain for $e = (n,s)$ **do**
13:           $l(e), u(e) \leftarrow$ Apply next estimator for $e$
14:        $\tilde{g}_u \leftarrow g_u(n) + u(e)$
15:        **if** $\tilde{g}_u < g_u(s)$ **and** $\tilde{g}_u \leq u_{prune}$ **then**
16:           $g_u(s) \leftarrow \tilde{g}_u$
17:           **if** $s$ in OPEN **then**
18:              Remove $s$ from OPEN
19:              Insert $s$ into OPEN with key $g_u(s)$ and parent $n$
20: **return** $\emptyset, \infty$

the while statement (Line 12) ends. Then, ordinary duplicate detection is performed in Lines 14–19.

We emphasize that in case the while loop (Line 12) terminates due to $g_u(n) + l(e) \geq g_u(s)$ being satisfied, then necessarily $\tilde{g}_u = g_u(n) + u(e) \geq g_u(s)$ will be satisfied as well and the path will be (justifiably) pruned. On the other hand, notice that $u(e)$ cannot be used in place of $l(e)$ for early pruning, since upper bound estimates tighten towards lower values. See Example 2 for a demonstration of using BEAST in its base setting.

**Example 2.** *Consider calling* BEAST *with* $u_{prune} = \infty$ *(i.e., base setting) on* $P$ *from Example 1. Tracing its run, at the first iteration of the outer while loop* $v_0$ *is removed from OPEN,* $\theta^1_{e_{01}}, \theta^1_{e_{02}}$ *and* $\theta^2_{e_{02}}$ *are invoked, and* $v_1, v_2$ *are inserted to OPEN with keys* $4, 5$*. At the second iteration* $v_1$ *is removed from OPEN,* $\theta^1_{e_{14}}, \theta^2_{e_{14}}$ *are invoked, and* $v_4$ *is inserted to OPEN with key* $10$*. At the third iteration* $v_2$ *is removed from OPEN,* $\theta^1_{e_{23}}, \theta^2_{e_{23}}$ *and* $\theta^1_{e_{24}}$ *are invoked, and* $v_3$ *is inserted to OPEN with key* $13$*. At the forth iteration* $v_4$ *is removed from OPEN and* BEAST *returns* $\langle e_{01}, e_{14} \rangle, 10$*.*

**Remark 1.** *In its base setting* BEAST *eventually uses the best estimates for edges leading to new nodes, thus it can be modified to directly jump to the best estimates in these cases. This was left out for simplicity of the pseudo-code.*

**Enhanced Setting.** We now consider the enhanced setting where $u_{prune}$ is set to some constant value (not $\infty$). In this setting $u_{prune}$ serves as an upper threshold that limits the search, similarly to *bounded cost search* (Stern et al. 2014). This manifests in two aspects. First, $u_{prune}$ is used as an

upper threshold to exit the while loop (Line 12) and avoid activating more expensive estimators in case the tight upper bound to $s$ is determined to be greater than $u_{prune}$. This is done in Line 12 where the condition $g_u(n) + l(e) \leq u_{prune}$ is tested, and if not fulfilled it breaks the loop. As explained in the base setting, $g_u(n) + l(e)$ serves as a lower bound to the tight upper bound to $s$ and can thus facilitate early stopping. Second, $u_{prune}$ is used as an upper threshold to prune (and not add to OPEN) any node with upper bound $> u_{prune}$. This is done in Line 15.

The primary purpose of using $u_{prune}$ with $U^* \leq u_{prune} < \infty$ is to avoid applications of redundant (and potentially expensive-to-compute) estimators. Additionally, it can decrease the size of OPEN, which implies less insertion operations and cheaper insert/delete operations. Since $U^*$ is unknown, setting this hyper-parameter to a meaningful value requires prior information. Practically, such information can be achieved by obtaining a suboptimal solution with $u_{sub} \geq U^*$, and using it to set $u_{prune} = u_{sub}$.

Finally, in case $U^* \leq u_{prune} < \infty$ is satisfied then a solution path $\pi$ will be found (if a solution exists) and it will be returned together with its tight upper bound $g_u(n) = U^*$ (Thm. 3). On the other hand, in case $u_{prune} < U^*$ is satisfied then $\emptyset, \infty$ are returned (Thm. 4). See Example 3 for a demonstration of using BEAST in its enhanced setting.

**Example 3.** *Consider calling* BEAST *with* $u_{prune} = 4$ *(i.e., enhanced setting,* $u_{prune} < U^* = 10$*) on* $P$ *from Example 1. Tracing its run, at the first iteration of the outer while loop* $v_0$ *is removed from OPEN,* $\theta^1_{e_{01}}, \theta^1_{e_{02}}$ *and* $\theta^2_{e_{02}}$ *are invoked, and* $v_1$ *is inserted to OPEN with key* $4$*. At the second iteration* $v_1$ *is removed from OPEN,* $\theta^1_{e_{14}}$ *is invoked. At the third iteration OPEN is empty and* BEAST *returns* $\emptyset, \infty$*.*

*Now consider calling* BEAST *with* $u_{prune} = 11$ *(i.e., enhanced setting,* $u_{prune} \geq U^* = 10$*) on* $P$ *from Example 1. Its run is identical to the run from Example. 2, with the same output, except that at the third iteration* $\theta^2_{e_{23}}$ *is not invoked.*

Next, we provide the theoretical guarantees for BEAST.

**Theorem 3** (Conditional Completeness and Optimality, Prob. 3)**.** BEAST, *with* $u_{prune} \geq U^*$*, returns a shortest path tightest upper bound* $\pi$ *and* $U^*$*, if a solution exists for* $P$*. In case no solution exists,* BEAST *returns* $\emptyset, \infty$*.*

*Proof.* First, it is straightforward to see that every node encountered by BEAST after the initial node is a successor of another node encountered during the search, as new nodes are only introduced in Line 8. Additionally, every node inserted into OPEN (except the initial node) is saved with a pointer to its parent node. Hence, whenever a goal node is found (at Line 5), it can necessarily be traced back to the initial node via a series of connected nodes, i.e., a valid solution path is returned at Line 6.

Second, BEAST inspects nodes that are removed from OPEN by best-first order w.r.t. upper bound of path cost. Since finding a goal node at Line 5 terminates the search, it is assured that the path leading to the first goal node found will be returned. Hence, the solution returned necessarily has the best (lowest) upper bound of path cost out of all the paths inspected by BEAST.

It remains to be shown that the search is systematic, namely that every path with tight upper bound up to $u_{prune}$ is inspected by BEAST; and that every edge encountered during the search is either tightly (fully) estimated, or it is at least estimated in a manner that enables BEAST to determine that it is not part of the solution path.

Consider any successor $s$ of a node $n$ that is popped from OPEN. If the node $s$ is encountered for the first time, then at first $g_u(s) \leftarrow \infty$ is set at Line 10, so the condition $g_u(n) + l(e) < g_u(s)$ at Line 12 is never satisfied. This means that the edge $e$ leading to $s$ will either be tightly (fully) estimated in case the current path to $s$ has tight upper bound smaller or equal to $u_{prune}$, or otherwise the tight upper bound to $s$ is greater than $u_{prune}$. Since $u_{prune} \geq U^*$ is assumed, this means that this path is not relevant and can be safely ignored. Indeed, in this case it is rightfully pruned in Line 15. If the node $s$ was already encountered earlier in the search, then the same mechanism described above applies, but additionally we have to consider the condition $g_u(n) + l(e) < g_u(s)$ at Line 12. In case it is not satisfied then this means that a better path was already found to $s$ so it can be safely ignored, and indeed it is pruned in Line 15.

Lastly, since we have shown that the search up to tight upper bound $u_{prune}$ is systematic, and considering that each edge has a finite number of estimators where each of them has finite run-time, BEAST necessarily terminates in finite time either when an optimal solution is found and returned (Line 6) with $U^*$, or when the search is exhausted up to tight upper bound $u_{prune}$ and BEAST reports that no solution exists (Line 20). Note that in the latter case the assumption $u_{prune} \geq U^*$ implies that $u_{prune}$ must have been set to $\infty$, so no solution at all exists. Hence, if $u_{prune} \geq U^*$ holds, BEAST is complete and returns an optimal solution. $\square$

**Theorem 4** (Soundness, Prob. 3). *For any value of $u_{prune}$, if $\emptyset, \infty$ are returned by* BEAST *then no solution exists for $P$ with tight upper bound that is smaller or equal to $u_{prune}$. Conversely, if* BEAST *returns a solution then it is correct.*

*Proof.* Following the proof of Thm. 3, since the search is systematic up to tight upper bound $u_{prune}$, it follows that if $u_{prune} < U^*$ holds then all paths with tight upper bound greater than $u_{prune}$ will be pruned, and since every solution has at least tight upper bound $U^*$, the search will necessarily terminate with $\emptyset, \infty$. The other direction, i.e., $u_{prune} \geq U^*$, is assured by Thm. 3. $\square$

### 4.2 The Second Algorithm: BEAUTY&BEAST

Algorithm 2 receives a TASP problem instance, and returns an optimal solution for it (Thm. 5). It works by first calling BEAUTY on $P$ (Line 1). If no solution is found then by the completeness of BEAUTY no solution at all exists, and $\emptyset, \infty$ are returned (Lines 2–3). Otherwise, the tightest upper bound of the solution found by BEAUTY, $u(\pi_{SLB})$, is obtained (Line 4). In case $L^* = u(\pi_{SLB})$ then necessarily $L^* = C^* = U^*$ and then $\pi_{SLB}$ and $\mathcal{B}^* = 1$ are returned (Line 5–6). Otherwise, BEAST is called on $P$ (Line 7) with $u_{prune} = u(\pi_{SLB})$, which is necessarily greater or equal to $U^*$, and thus by Thm. 3 a shortest path tightest upper bound

---

Algorithm 2: BEAUTY&BEAST
___
**Input**: Problem $P = (G, v_s, V_g)$, where $G$ is an estimated weighted digraph with cost estimators function $\Theta$
**Output**: Path $\pi^*$, bound $\mathcal{B}^*$
 1: $\pi_{SLB}, L^* \leftarrow$ Solve SLB for $P$ using BEAUTY
 2: **if** $\pi_{SLB} = \emptyset$ **then**
 3:     **return** $\emptyset, \infty$
 4: $u(\pi_{SLB}) \leftarrow u_{\Theta(\pi_{SLB})}$  // Get the upper bound of $\pi_{SLB}$
 5: **if** $L^* = u(\pi_{SLB})$ **then**
 6:     **return** $\pi_{SLB}, 1$
 7: $\pi^*, U^* \leftarrow$ Solve SUB for $P$ using BEAST, $u(\pi_{SLB})$
 8: **if** $L^* = 0$ **and** $U^* > 0$ **then**
 9:     **return** $\pi^*, \infty$
10: **return** $\pi^*, U^*/L^*$

---

is returned. If $U^* > L^* = 0$ then the shortest path tightest upper bound that was found, $\pi^*$, is returned together with $\infty$ (Line 8–9). Otherwise $\pi^*$ and $\mathcal{B}^* = U^*/L^*$ are returned.

**Remark 2.** *Note that in Alg. 2* BEAUTY *can be replaced by any algorithm that solves SLB optimally. Moreover, it can be replaced by an anytime algorithm, and then each time the lower bound or upper bound are improved (tightened) they can be translated to a tightened $\mathcal{B}$.*

**Theorem 5** (Completeness and Optimality Prob. 1). BEAUTY&BEAST *is complete. Furthermore, if a solution exists for $P$ then a tightest admissible shortest path $\pi$ and $\mathcal{B}^*$ are returned (if $U^* > L^* = 0$ holds, then $\mathcal{B}^* = \infty$).*

The proof follows directly from the completeness and optimality of BEAUTY and BEAST, and from Thm. 2.

## 5 Empirical Evaluation

The theoretical guarantees of BEAST (base setting) and BEAUTY&BEAST (enhanced setting) assure optimality and completeness, but do not provide information about their runtime performance. We therefore empirically evaluate the algorithms in diverse settings, based on AI planning benchmark problems that were modified to have multiple action-cost estimators, so that these induce TASP problems.

The set of problems was taken from a collection of IPC (International Planning Competition) benchmark instances[1]. Starting from the full collection, we selected all domains that use action costs, were part of an optimal track in IPC, and without duplication (e.g., only one version of the Elevators domain). Then, we created additional problems by using different configurations of costs. Lastly, for all domains and problems, we synthesized three estimators, using six parameters $f_1, ..., f_6$, according to the scheme described below.

The original cost $c_{old}(e)$ (that is implied by the original domain and problem files) of each edge $e$ was mapped to a new cost $c_{new}(e)$ that satisfies

$$c_{old}(e) \times f_3 \leq c_{new}(e) \leq c_{old}(e) \times f_4, \quad (11)$$

i.e., the exact value of $c_{new}(e)$ is contained in the interval $[c_{old}(e) \times f_3, c_{old}(e) \times f_4]$. Lower bounds were defined by

$$l_e^1 := c_{old} \times f_1, l_e^2 := c_{old} \times f_2, l_e^3 := c_{old} \times f_3, \quad (12)$$

---

[1]See https://github.com/aibasel/downward-benchmarks.

| Domain | Instances | $\theta^{max}$ Reduction | Extra $\theta^{max}$ Reduction | Pruned Nodes | $\mathcal{B}^*$ |
|---|---|---|---|---|---|
| Barman | 135 | 39.46±3.22 | 7.00 ± 9.03 | 2.20 ± 2.80 | 1.49±0.12 |
| Caldera | 135 | 17.91±2.62 | 27.16± 5.20 | 8.28 ± 1.49 | 1.52±0.13 |
| Elevators | 135 | 70.79±4.14 | 69.24±24.73 | 30.34±15.07 | 1.56±0.26 |
| Settlers | 81 | 36.08±3.73 | 33.31± 8.77 | 16.26± 2.48 | 1.52±0.13 |
| Sokoban | 135 | 44.55±6.73 | 17.69±25.95 | 5.47 ± 8.31 | 1.54±0.21 |
| Tetris | 135 | 36.52±4.62 | 28.81±12.32 | 14.82± 7.21 | 1.54±0.20 |
| Transport | 135 | 50.55±3.36 | 62.40±19.79 | 17.95± 5.47 | 1.52±0.15 |
| **All domains (avg±std)** | **891** | **42.64±15.88** | **35.08±27.69** | **13.40±11.75** | **1.53±0.18** |
| **All domains (min–max)** | **891** | **14.28–79.43** | **0.07–98.37** | **0.02–60.12** | **1.03–2.47** |

Table 1: Summarized performance data of BEAST ($\infty$), BEAST ($u(\pi_{SLB})$), with breakdown by domains. $\theta^{max}$ is the number of expensive estimators. Column 3 refers to $1 - (\theta^{max}(\text{BEAST } (\infty))/\theta^{max}(\text{El-UCS}))$, Column 4 refers to $1 - (\theta^{max}(\text{BEAST } (u(\pi_{SLB})))/\theta^{max}(\text{BEAST } (\infty)))$, Column 5 refers to pruned nodes out of generated nodes for BEAST ($u(\pi_{SLB})$), Column 6 refers to $\mathcal{B}^*$. The entries for each domain (Rows 2–8) show average $\pm$ standard deviation. Cumulative results show average $\pm$ standard deviation (Row 9) and minimum–maximum (Row 10). All results are in percentages, except $\mathcal{B}^*$ values.

with $f_3 \geq f_2 \geq f_1 \geq 1$, so that $l_e^i$ is the i$^{th}$ lower bound ($l_e^1$ is the loosest and $l_e^3$ is the tightest). Upper bounds were analogously defined by

$$u_e^1 := c_{old} \times f_6, u_e^2 := c_{old} \times f_5, u_e^3 := c_{old} \times f_4, \quad (13)$$

with $f_6 \geq f_5 \geq f_4 \geq f_3$, so that $u_e^i$ is the i$^{th}$ upper bound ($u_e^1$ is the loosest and $u_e^3$ is the tightest).

To diversify the estimator sets for different edges, the parameters for lower bounds $f_1, f_2, f_3$ were taken from the sets

$$f_1 \in \{1, 2, 3\}, f_2 \in \{f_1, f_1 + 1, f_1 + 2\}, \\ f_3 \in \{f_2, f_2 + 1, f_2 + 2\}. \quad (14)$$

Similarly the parameters for upper bounds $f_4, f_5, f_6$ were taken from the sets

$$f_4 \in \{f_3 + 1, f_3 + 2, f_3 + 3\}, \\ f_5 \in \{f_4, f_4 + 1, f_4 + 2\}, \quad (15) \\ f_6 \in \{f_5, f_5 + 1, f_5 + 2\}.$$

This induced a very wide range of relations between the different estimators and a wide variety of uncertainty levels (reflected by $\mathcal{B}^*$). The choice of configuration was taken according to the result of a simple hash function, that depends on $c_{old}(e)$ and a user-input seed, described as follows:

$$\text{Hash} = (c_{old}(e) + \text{seed}) \mod 27, \quad (16)$$

so that every value of Hash corresponded to one estimator configuration for the lower and upper bounds. Each problem was run once per seed, where all seeds from the set $[0, 26]$ were taken (namely, 27 seeds per problem). The full list of domains, problems and configurations that were used in the experiments is detailed in (link redacted for anonymity).

We note that the estimator configurations that were chosen according to the hash function of Eq. (16) guarantee that the same ground action, in different states, will have the same cost estimates.

BEAST and BEAUTY&BEAST were implemented as search algorithms in *PlanDEM* (Planning with Dynamically Estimated Action Models (Weiss and Kaminka 2023a), an open source C++ planner that extends Fast Downward (FD) (Helmert 2006) (v20.06). All experiments were run on an Intel i7-1165G7 CPU (2.8GHz), with 32GB of RAM, in Linux. We also implemented *Estimation-time Indifferent* UCS (El-UCS), a UCS algorithm that uses the most accurate estimate on each edge it encounters, to serve as a baseline for solving SUB.

We measured the performance of solving SUB via El-UCS, BEAST (base setting) and BEAUTY&BEAST (enhanced setting with information obtained from solving first SLB). We report the results (summarized in Table 1, extended in Fig. 2) from problem instances which all algorithms solved successfully, i.e., found optimal solutions, within 5 minutes. Overall, we report on a cumulative set of 891 problem instances, spanning 7 unique domains.

**Base Setting vs. El-UCS** In its base setting BEAST expands the same nodes as El-UCS, but with potentially fewer expensive estimates (replaced by less expensive ones). Thus, we are interested in the relative savings that it achieves in practice. We denote the number of the *most expensive* estimators invoked during the search of an algorithm $ALG$ by $\theta^{max}(ALG)$. Column 3 in Table 1 provides the relevant data: it shows $1 - (\theta^{max}(\text{BEAST } (\infty))/\theta^{max}(\text{El-UCS}))$ in percentages, per domain (Rows 2–8), cumulatively by average$\pm$standard deviation (Row 9), and by range minimum–maximum (Row 10).

Roughly 40% of the expensive estimators are saved on average, with large differences across domains and problems. Over all, the range is from 14% to almost 80%. The top left plot in Fig. 2 shows the full empirical distribution (histogram), illustrating the high variability in the results. We suspect that different distributions of edge costs explain some of the variability, a topic left for future research.

**Enhanced Setting vs. Base Setting** We can test the performance boost that can be achieved in practice from using BEAST in its enhanced setting with $u_{prune} = u(\pi_{SLB})$, by running BEAUTY&BEAST. Column 4 in Table 1 provides this data: the format is similar to that of Column 3, but instead it shows $1 - (\theta^{max}(\text{BEAST } (u(\pi_{SLB})))/\theta^{max}(\text{BEAST } (\infty)))$. It can be seen that roughly 35% of the expensive estimators are saved on average *w.r.t. to the base setting of* BEAST, again with very

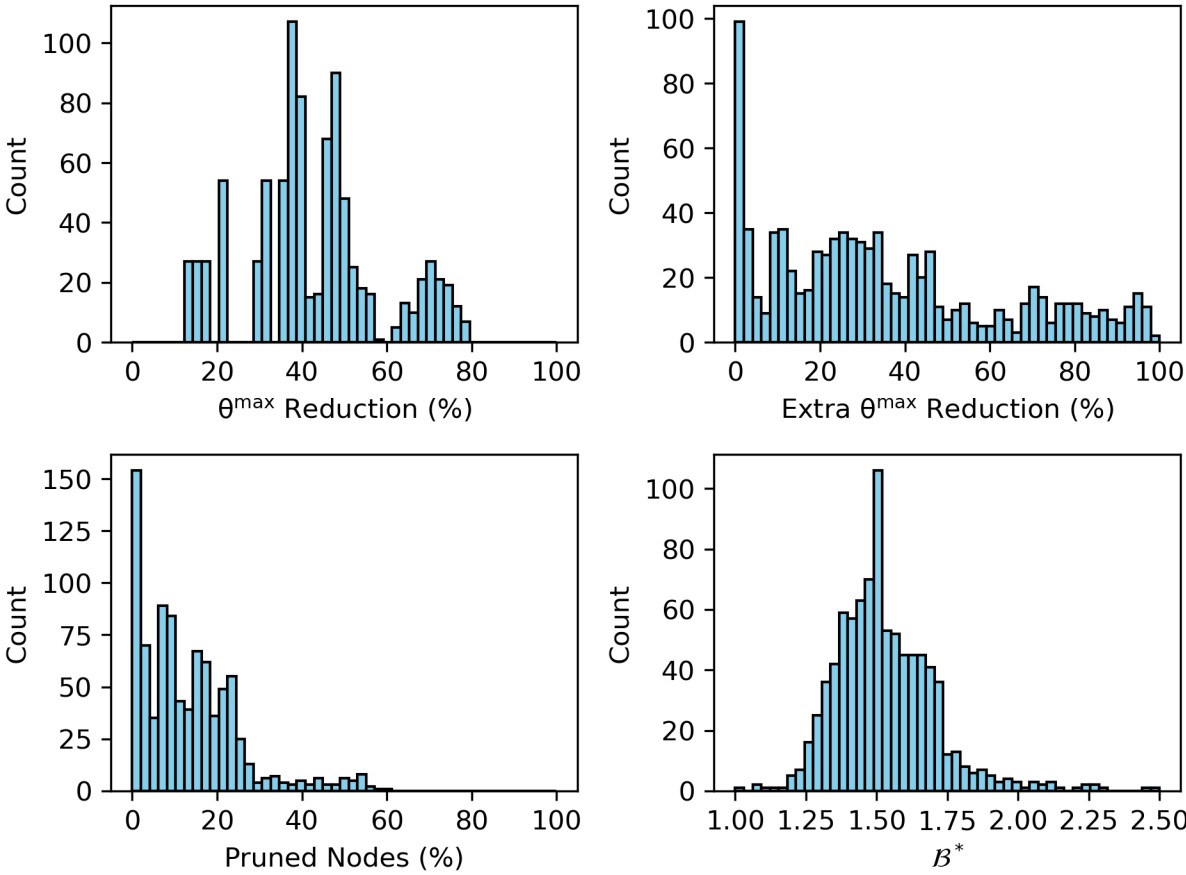

Figure 2: Histograms of $1 - (\theta^{max}(\text{BEAST } (\infty))/\theta^{max}(\text{El-UCS}))$ [top left], $1 - (\theta^{max}(\text{BEAST } (u(\pi_{SLB})))/\theta^{max}(\text{BEAST } (\infty)))$ [top right], pruned nodes percentage for BEAST $(u(\pi_{SLB}))$ [bottom left], $\mathcal{B}^*$ [bottom right], based on all domains.

large differences across domains and problems, where over all problem instances the range is from 0% to around 98%. The top right plot in Fig. 2 shows the full empirical distribution, revealing that in the most common case the savings are rather low, but on the other hand in quite a few cases the savings achieved are very substantial.

Column 5 in Table 1 presents pruned nodes (out of generated nodes) for BEAST $(u(\pi_{SLB}))$, and it shows that roughly 15% of the generated nodes are pruned on average. Column 6 shows $\mathcal{B}^*$ (which is a property of the problem instance), where we can see that approximately it was on average 1.5 (the values in this column are not in percentages).

For both of these measures, we see high variability in the results: The range of pruned nodes (Column 5) is from 0% to 60%. The range of $\mathcal{B}^*$ over all problems was from 1 to 2.5. However, examining the distributions for both columns (Fig. 2; bottom left plot for pruned nodes, bottom right for $\mathcal{B}^*$), we see that the distributions are qualitatively very different. We will investigate this in future research.

We conclude that BEAST seems to offer significant empirical gains over El-UCS, and that using the information obtained from solving SLB can, though not always, provide an additional significant performance boost.

## 6 Conclusions

This paper introduces—and offers a solution to—the *tightest admissible shortest path* (TASP) problem. It determines how close can one get to cost-optimality, given bounded edge-weight uncertainty. The formalization relies on a recently suggested generalized framework for *estimated weighted directed graphs*, where the cost of each edge can be estimated by multiple estimators, where every estimator has its own run-time, and returns lower and upper bounds on the edge weight. We show how to generally solve TASP problems by reducing it to the solution of two more basic problems—SLB and SUB. We then present a complete algorithm that obtains optimal solutions for TASP problems, which uses a coupling between SLB and SUB to reduce overall run-time compared to separately solving SLB and SUB. Experiments support the efficacy of the approach.

There are many directions for future research. Algorithmic extensions to informed search seem highly relevant, as well as exploring additional trade-offs between search and estimation time to reduce overall run-time. Extending the framework to utilize priors on estimation times to choose estimators across edges also appear promising.

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
