# OpenReview forum: "Tightest Admissible Shortest Path"
_icaps-conference.org/ICAPS/2024/Conference — ICAPS 2024_

### Official Review · Reviewer_PZXt · 2024-01-10

**Significance And Importance:** 2
**Soundness:** 4
**Novelty:** 2
**Clarity:** 4
**Overall Evaluation:** 1
**Confidence:** 3

**Weaknesses:**

2: No major or minor weaknesses.

**Contributions Of The Paper:**

The paper introduces a new problem - computing TASP on Estimated Weighted Digraph. It proposes a straightforward yet effective search-based method to address the problem and offers substantial theoretical analysis.

**Ethical Considerations:**

(1) Not Applicable: The paper does not have any ethical considerations to address

**Nomination For Best Paper:**

No

**Questions For Authors:**

Could you provide a concrete example of an EWDG in a planning problem and illustrate how TASP can effectively solve it, while highlighting why other existing methods are not suitable?

**Reproducibility:**

3: Authors describe the implementation and domains in sufficient detail.

**Strengths Of The Paper:**

The paper is well-written and easy to follow, and it reveals an interesting connection between SLB, SUB and TASP.

**Weaknesses Of The Paper:**

My concerns primarily stem from novelty and impact.
The paper studies a variant of a problem that appears in an existing literature (Weiss, et al., ECAI2023). Most of its components are derived from this with minor modifications, including definitions, theorems, and the SLB algorithm.
However, it could be more significant if this following work can explore in directions like:

1. Application: Despite a paragraph explaining the motivation behind studying this problem (TASP on EWDG), a concrete real-world
   example of its application is absent;

2. Theoretical insight on estimators: The impact is questionable due to the strong assumptions made about estimators,
i.e., their behaviour must be predictable (more time yields tighter bound), which seems odd when we're dealing with a
problem that involves uncertainty. Also note that the difficulty of obtaining lower and upper bounds may differ; e.g., for an edge in a road network, the lower bound could simply be the Euclidean distance, but determining the upper bound may require simulation;

3. Better search framework for TASP, rather than trivially call SLB then SUB in algorithm 2.
It could be more interesting to elaborate on 'remark 2', e.g., using cheap estimators as a heuristic, or interleaving SLB and SUB.

As a result, I regard it as an incremental work with limited novelty and small impact.

I believe it is crucial to provide a concrete example of TASP on EWDG to substantiate the significance of this problem. It is also important to incorporate a domain in the
experiment, with real estimators (such as those from the concrete example) to demonstrate actual performance in terms of elapsed time.

I am open to revising my opinion based on the response.
---
Revision:
The information provided by the authors has convinced me of the impact of this work. Thus, I have adjusted my decision to 'weak accept', even though I remain not fully satisfied with its novelty.

---

> ### Author Rebuttal · Authors · 2024-01-27
>
> There have been quite a few papers on different aspects of gradual estimation in planning and search. These include works on planning with semantic attachments, that focus on integration of external procedures in domain-independent planning, with cheap/expensive alternatives; and a variety of papers on lazy search methods in motion planning, dealing with expensive edge-existence verification procedures. In our paper we extend the EWDG framework for dealing with similar applications, where the focus is on cost estimation alternatives. Hence, developments in this direction could impact many interesting real-world problems.
>
> Concretely, please see the application described in the rebuttal to reviewer noiS (“To all reviewers”). Here, upper bounds on EC are crucial, lower bounds are also useful, and thus TASP becomes very relevant. Previous work on EWDG (BEAUTY) does not tackle it, other works do not account for multiple estimators for edge weights, and ignoring cheap estimators (EI-UCS) is wasteful. So, this is a concrete example for EWDG, that TASP effectively addresses, while other methods (inc. BEAUTY) don’t.
>
> An experiment with real estimators is interesting but would require a lot of engineering as it involves physical platforms. However, we can provide positive results from our experiments on actual performance in terms of elapsed time – please see the answer to Q2 of Reviewer 6irk.
>
> Regarding novelty, it is difficult to argue with a subjective claim of “minor modifications”. The paper extends the definition of admissibility (Def. 4), introduces two new problems (SUB and TASP), connects them to SLB (Thm. 2), provides theoretically proven optimal algorithms for them (Algs. 1,2, Thms. 3-5), and demonstrates empirical superiority over the existing alternative (Table 1). Note that BEAST differs from BEAUTY in ideas, proofs and behavior. Additionally, we agree that perhaps better TASP algorithms exist – but Alg 2 would be the baseline against which they are compared, as it is the first. And we strongly believe that before exploring more TASP algorithms it is better to first lay solid foundations.
>
> Lastly, the EWDG framework and our algorithms are actually rather flexible so they can account for diverse use cases and settings. E.g., one can modify Def. 1 so that estimators monotonicity is only w.r.t. expected run time, and SLB, SUB and TASP would still make sense, while BEAUTY and BEAST will perform similarly on average.

---

### Official Review · Reviewer_6irk · 2024-01-21

**Significance And Importance:** 2
**Soundness:** 2
**Novelty:** 3
**Clarity:** 4
**Overall Evaluation:** 1
**Confidence:** 3

**Weaknesses:**

0: Minor weaknesses requiring some work to be addressed for the paper to be accepted.

**Contributions Of The Paper:**

The paper introduces the tightest admissible shortest path problem. Here, the input network where edge costs are not given explicitly but lower and upper bound cost oracles can be queried. Tighter bounds require more query time and exact costs might not be available at all. The optimization problem asks for the best approximation to the real shortest path costs that can be obtained in this setting. To find a solution, the problem is split into retrieving the tightest lower and upper bound separately. For the lower bound, there already existed an approach presented in previous work. To complement this algorithm, an upper bound algorithm is designed here. It is further explained how to integrate both algorithms into a unified one with improved performance. The experimental evaluation demonstrates the efficiency of the proposed  methods.

**Ethical Considerations:**

(1) Not Applicable: The paper does not have any ethical considerations to address

**Nomination For Best Paper:**

No

**Questions For Authors:**

Q1. What are the theoretical worst-case running times of the BEAST and the combined algorithm? Are there scenarios where computing the best cost estimate for all edges directly and then proceeding with classical search algorithms is more efficient?

Q2. What are the running times? How do they compare to the baseline? What would the oracle cost difference need to be to get a significant running time reduction with the new approach?

**Reproducibility:**

5: Code and domains (whichever apply) are already publicly available

**Strengths Of The Paper:**

S1. The proposed problem is interesting and comes with real-world applications.

S2. The BEAST algorithm and its combination with the BEAUTY algorithm are sensible and relevant properties are proven.

**Weaknesses Of The Paper:**

W1. There is no running time analysis.

W2. The experiments do not consider a real application (although one is explained in the introduction), but instead artificial cost estimates are used. Therefore, there are also no cost differences for the estimators that can be taken into account. This somewhat defeats the purpose of the whole approach, where the total running time includes this aspect. It is only measured how many expensive estimator costs are saved. But this is not the full story, as asking several cheaper estimation queries could still accumulate a higher running time in total than going directly for a more precise estimate. The total number of oracle calls for each precision level are not presented, and running times are not provided at all.

W3. Minor comment: The interpretability of the histograms might be increased by going with percentages instead of absolute counts.

---

> ### Author Rebuttal · Authors · 2024-01-27
>
> Q1: The complexity analysis of Alg. 1 (BEAST) follows the analysis of Uniform Cost Search (UCS). We denote: b is the largest branching factor in G; k_max is the cardinality of the largest estimator sequence in Theta; and epsilon_u is the minimum tight upper bound for an edge cost, out of all the estimators in Theta.
>
> The worst-case time complexity of Alg. 1 is O(b^{1+floor(U*/epsilon_u)}k_max).
>
> Namely, U* and epsilon_u replace the regular C* and epsilon in the complexity of UCS, and this is multiplied by k_max.
>
> Similarly, the worst-case time complexity of BEAUTY is O(b^{1+floor(L*/epsilon_l)}k_max), where epsilon_l is defined analogously for the minimum tight lower bound.
>
> Finally, the worst-case time complexity of Alg. 2 is O((b^{1+floor(U*/epsilon_u)}+b^{1+floor(L*/epsilon_l)})k_max).
>
> I.e., it is a union of the running times of BEAUTY and BEAST.
>
> This analysis assumes that every estimator has O(1) runtime. Otherwise, the expressions need to change according to the assumptions on the running time of the estimators.
>
> We note that the space complexities of the algorithms are similar, but without the k_max factor.
>
> We didn’t include these results due to lack of space, but we will add them in the final version if more space is made available.
>
> A scenario where EI-UCS is more efficient than BEAST is when the average running times of all estimators are roughly the same. Then, it does not matter which estimators are invoked, but only how many are invoked. However, in this case there is no point to even consider using a gradual estimation framework. The answer to Q2 sheds more light.
>
> Q2: The running times in actual applications are highly dependent on the estimation times. In our experiments we abstracted them, so the running time only accounts for pure search. Since EI-UCS and BEAST expand the same nodes (as explained in the paper), their search time is almost identical, and thus not informative.
>
> Nonetheless, we can provide a conservative rule of thumb based on our empirical evidence. Denote tau^i as the average estimation time for the ith estimator. Then:
>
> BEAST performs better than EI-UCS when:
> * k_max=2 and 2*tau^1<tau^2, and
> * k_max=3 and 3*tau^2<tau^3.
> Runtime reduction grows monotonically with higher ratios of tau^(k_max) to other tau^i. For instance, if tau^(k_max) is an order of magnitude greater than other tau^i, a very significant time reduction is expected.
>
> We will add in the final version an analysis of expected run time gains of BEAST.

---

### Official Review · Reviewer_noiS · 2024-01-22

**Significance And Importance:** 2
**Soundness:** 3
**Novelty:** 3
**Clarity:** 4
**Overall Evaluation:** 2
**Confidence:** 4

**Weaknesses:**

1: Minor weaknesses that are easily fixable.

**Contributions Of The Paper:**

The paper introduces the "tightest admissible shortest path" problem and provides an optimal algorithm for solving it. The authors present initial empirical results demonstrating their algorithm out performing two simpler baselines.

**Ethical Considerations:**

(1) Not Applicable: The paper does not have any ethical considerations to address

**Nomination For Best Paper:**

No

**Questions For Authors:**

Just a few questions:

1) You cite that the seeding of the cost estimators yielded the same cost estimates for the same ground action in different states. I could easily see this not being the case (especially in something like robotics where state geometry may play a large role), is that a property that is strictly necessary to maintain the results of your proofs?

2) Why did you choose to normalize the BEAUTY&BEAST estimator against just BEAST and not EI-UCS like in the previous section? It seems like a missed continuity opportunity?

3) Admittedly, it's difficult to come up with a set of monotonically constraining cost estimators for a set of synthetic domains, and I have no issue with your choice there. However, could anything be said about how these synthetic cost functions performed against each other? It's "unlikely" but it could be the case that f1=f2=f3 and f4=f5=f6? Is the possible lack of variation in the final functions giving an unfair advantage to BEAST and BEAUTY&BEAST?

**Reproducibility:**

4: Authors promise to release code and domains (whichever apply).

**Strengths Of The Paper:**

First, the algorithm name is splendid, a really great extension of the previous work (not a real valid research strength, but I had to applaud this).

The paper is really quite clearly written, it was very easy to follow the thought process leading up the the BEAUTY&BEAST algorithm. The descriptions of the algorithms leveraged were very helpful in getting up to speed and setting the context for the paper. The proofs are well written and I believe them to be correct. I think the results presented a good pairing of baseline algorithms to compare against. Overall, a very nice paper.

**Weaknesses Of The Paper:**

I found only two "weaknesses", which are more like "what I wish had been extended a bit further".

The first is the motivation for the new domain. The example of Google Maps is okay, but I wonder if there is a better motivating example, or just additional examples. It's not immediately obvious to me where this may arise in practice. Fleshing this out would definitely strengthen the paper in my opinion.

The paper is so well written that when I got to the results section, I felt a little blindsided by a lot of synthetic complexity to create these new kinds of problems based on the IPC domains. Perhaps this is a restatement of the first thing I mentioned, but I did not get an intuitive idea of the utility of multiple edge cost estimators based on the chosen benchmark set.

The results being presented, seem like a useful improvement over the baselines in terms of reduction of calls to the most expensive edge cost evaluator. It's not super clear how impactful that is though. Whereas in robotics domains where edge cost/feasibility is expensive collision checking, that can have a practical reportable improvement in terms of calls to the collision checker and runtime. I wish this kind of analysis had been done to really sell the win here.

---

> ### Author Rebuttal · Authors · 2024-01-27
>
> To all reviewers: We describe another practical application which will be added to the paper.
>
> One of the major sources of uncertainty in planning multi-segment flight routes for logistics drone missions is energy consumption (EC), which is affected by winds, weight, and other factors. Optimistic and pessimistic EC bounds can be very useful for better drone scheduling, e.g., by using the bounds to devise greedy/conservative schemes [1]. Based on wind vector predictions [2] it’s possible to estimate EC bounds in a flight segment (graph edge) in multiple ways [1,2,3]. These may involve a mix of online queries and computation, and can be used to obtain L*, U*, and B*. Here B* can be interpreted as the maximum energetic overhead for choosing a conservative route, or if L* and U* are obtained for the same route, then B* quantifies its EC uncertainty.
> To sum up:
> * EC uncertainty in drones is a real issue, thus bounds on EC are useful for mission planning.
> * Estimation of EC bounds is possible by various methods. This is an active area of research.
> * For each flight-segment (edge) we may use multiple methods, varying in computational cost.
>
> [1] Drone scheduling model for delivering small parcels to remote islands considering wind direction and speed, Computers & Industrial Engineering, 2022
>
> [2] Real-time Wind Predictions for Safe Drone Flights in Toronto, Results in Engineering, 2022
>
> [3] A procedure for power consumption estimation of multi-rotor unmanned aerial vehicle, Journal of Physics: Conference Series, 2020
>
> Analogous examples exist for robots and unmanned vehicles, for a variety of measures of interest (travel time/pollution output/energy consumption) that depend on the medium traversed. There are many methods for estimating those, depending on the application, required precision, and available computational resources: from relatively simple engineering formulas, to physics-based simulators.
>
> To Reviewer noiS:
>
> Q1: The theoretical properties of our algorithms hold also in the case of state-dependent costs. Will clarify.
>
> Q2: We thought it would be mostly interesting to see the improvement of the enhanced version vs. the basic version of BEAST (which itself outperforms EI-UCS), but we can indeed add a comparison to EI-UCS.
>
> Q3: We did not observe any noticeable advantage for a specific seed configuration or specific synthetic cost function. Most of the correlation in performance we witnessed was related to the choices of domain and problem instance.

---

### Meta-Review · Area_Chair_DZGv · 2024-02-03

**Recommendation:** Accept (Poster)
**Confidence:** 4

**Metareview:**

A consensus to (weak) accept this paper was reached after reviewer PZXt increased their evaluation based on the authors' rebuttal.

We ask the authors to follow the reviewers' recommendations and include the needed clarifications in the final version of their paper.

**Ethical Considerations:**

(1) Not Applicable: The paper does not have any ethical considerations to address